# Comparative Performance Analysis of the DC-AC Converter Control System Based on Linear Robust or Nonlinear PCH Controllers and Reinforcement Learning Agent

**DOI:** 10.3390/s22239535

**Published:** 2022-12-06

**Authors:** Marcel Nicola, Claudiu-Ionel Nicola

**Affiliations:** Research and Development Department, National Institute for Research, Development and Testing in Electrical Engineering—ICMET Craiova, 200746 Craiova, Romania

**Keywords:** robust control, Port Controlled Hamiltonian, Reinforcement learning, DC-AC converter, grid

## Abstract

Starting from the general topology and the main elements that connect a microgrid represented by a DC power source to the main grid, this article presents the performance of the control system of a DC-AC converter. The main elements of this topology are the voltage source inverter represented by a DC-AC converter and the network filters. The active Insulated Gate Bipolar Transistor (IGBT) or Metal–Oxide–Semiconductor Field-Effect Transistor (MOSFET) elements of the DC-AC converter are controlled by robust linear or nonlinear Port Controlled Hamiltonian (PCH) controllers. The outputs of these controllers are modulation indices which are inputs to a Pulse-Width Modulation (PWM) system that provides the switching signals for the active elements of the DC-AC converter. The purpose of the DC-AC converter control system is to maintain *u_d_* and *u_q_* voltages to the prescribed reference values where there is a variation of the three-phase load, which may be of balanced/unbalanced or nonlinear type. The controllers are classic PI, robust or nonlinear PCH, and their performance is improved by the use of a properly trained Reinforcement Learning-Twin Delayed Deep Deterministic Policy Gradient (RL-TD3) agent. The performance of the DC-AC converter control systems is compared using performance indices such as steady-state error, error ripple and Total Harmonic Distortion (THD) current value. Numerical simulations are performed in Matlab/Simulink and conclude the superior performance of the nonlinear PCH controller and the improvement of the performance of each controller presented by using an RL-TD3 agent, which provides correction signals to improve the performance of the DC-AC converter control systems when it is properly trained.

## 1. Introduction

Although there are various topologies and connection schemes for the connection of microgrids to the main grid, in general, it can be said that the central element is a voltage source inverter, represented by a DC-AC converter that can connect a DC source power to the main grid. Among the other important elements of the system, a special role is played by the connection filters attempting to perform primary filtering due to load fluctuations or parametric variation. In a top-down approach to the general issues that can be found in a microgrid, we can start with the issues of optimization and forecasting from an economic point of view [1,2] and then analyze the control elements of the main subassemblies of the microgrid, i.e., the DC-DC converter [3,4], DC-AC converter [5,6,7], battery energy storage system (BESS) [8,9], and last but not least specific connection elements in the case of electric vehicles connected to the microgrid [10]. 

When the purpose of such a system is to maintain certain quality quantities (e.g., *u_d_* and *u_q_* voltages described in the *d*-*q* reference frame) to prescribed values with minimal fluctuations when the load and system parameter values may vary, it is necessary to use high-performance controllers for DC-AC converter control. Traditionally, PI-type controllers are used, which offer relatively good performance and parametric robustness, but only around static operating points established after the tuning of the PI-type controller [11]. Naturally, to obtain superior control performances, a series of modern types of controllers have been developed and implemented specifically for the control of the main elements of the microgrid described above, including adaptive controllers [12], robust controllers [13,14,15,16,17] in case of significant parametric variations, neuro-fuzzy controllers [18], as well as nonlinear controllers based on the Passivity theory, including nonlinear PCH [19,20,21,22,23]. 

In terms of Machine Learning types, we can mention the RL-TD3 agent [24,25,26,27], which can improve the performance of the DC-AC converter control system. The RL-TD3 agent resembles the architecture of an industrial process control system through a very strong analogy in terms of information acquisition and command provision, as well as optimization of an overall quality index. After the phases of training and validation of an RL-TD3 agent, it provides correction signals to the command signals leading to optimized and increased performance of the control system. 

The microgrid topology discussed in this article and the control objectives are based on a benchmark presented in [16,17,22,23]. Thus, the performance of DC-AC converter control systems is compared when using PI-type, robust and PCH-type controllers. The performance indicators used are: steady-state error of *u_d_* voltage; error ripple of *u_d_* voltage; and THD current phase *a* of the microgrid-to-the-main-grid connection system using a DC-AC converter. Moreover, balanced/unbalanced or nonlinear loads are used for these comparisons of the performance of the mentioned control systems.

The main contributions of this paper can be summarized as follows:Presentation, synthesis, and implementation of the robust control algorithm for DC-AC converter control;Presentation, synthesis, and implementation of the PCH control algorithm based on the passivity theory for the DC-AC converter control;Presentation, synthesis, and implementation of an RL-TD3 agent, by covering the stages of creation, training, testing and validation for each of the PI, robust and PCH controllers;Implementation in Matlab/Simulink of the software applications for the calculation of the steady-state error performance indicators and the error ripple of the *u_d_* voltage and THD current phase a of the microgrid-to-the-main-grid connection system using a DC-AC converter for the comparative analysis of PI, robust and PCH control systems with or without the RL-TD3 agent.

The rest of the paper is structured as follows: Section 2 presents the robust control of the DC-AC converter and the Matlab/Simulink implementation of the robust controller, while the PCH-type control and the Matlab/Simulink implementation of the PCH-type controller are presented in Section 3. Section 4 presents the numerical simulations, and future works are presented in the final section.

## 2. Robust Control of the DC-AC Converter

In general, the coupling of a microgrid (considered as a DC power source in the structure discussed below) to the grid is achieved by means of a voltage source inverter (DC-AC converter). Assuming that the DC power source is capable of supplying a constant current to power the DC-AC converter, Figure 1 shows the block diagram for the DC-AC converter control system using a robust controller. 

The elements in the block diagram are shown in the *d*-*q* frame, and to synchronize the voltage at the output of the DC-AC converter with the voltage supplied by the grid, references *i_dref_*, *i_qref_* are set initially to 0, while the breaker is set to the closed position. The grid voltages are filtered by a low-pass filter to reduce harmonics and then supply a feed-forward to the robust controller outputs to obtain PWM modulation pulses for the DC-AC converter control.

The grid-characteristic currents *i_a_*, *i_b_*, *i_c_*, are dictated by the consumers connected to it and represent the input quantities for the robust controller, which will be synthesized using the robust systems theory. This controller will supply the control signals to a PWM generator, and by driving active MOSFET or IGBT elements in the DC-AC converter, *u_d_* voltage will be kept constant, which is the main objective of the control system for the presented benchmark. We specify that in the microgrid topology shown in Figure 1, there is no BESS precisely in order to follow the benchmark presented. From the point of view of the synthesis of the controllers proposed in this article, the absence or presence of a BESS does not influence the synthesis of these controllers or the performance of these control systems. This is due to the fact that in the currents *i_a_*, *i_b_*, *i_c,_* which represent inputs for the controller, there are fluctuations caused by consumers, and possible BESS’, both in the stationary regime and in dynamic regime, as a result of their connection or disconnection. Moreover, [8] presents the control of the main phenomena occurring when there is a BEES, namely their charging or discharging according to certain criteria imposed by the connection to the microgrid. These refer to the charging and discharging of the BESS when the voltage at its terminals is lower, respectively higher by a set percentage than the voltage which is intended to be kept constant in the microgrid. These goals are achieved through the use of classical PI-type cascade controllers, where the charging/discharging current of the BESS is regulated in the inner loop, and the voltage at the BESS terminals is regulated in the outer loop. 

### 2.1. Mathematical Description of the Robust Control for DC-AC Converter

In the *d*-*q* frame, for Figure 1, the quality quantities *u_d_* and *u_q_* voltages are defined in the sense that the purpose of the DC-AC converter control is to maintain the constant values of *u_d_* = 310 V and *u_q_* = 0 V. To use the concepts of the robust control systems theory, plant *G* is presented, starting from the single phase representation in Figure 2, where the notations are the usual ones.

Thus, the mathematical description takes the form given by Equations (1) and (2).
(1)$$\dot{x}=Ax+B_{1}w+B_{2}u$$
(2)$$y=e=C_{1}x+D_{1}w+D_{2}u$$
where: $x={\left[\begin{array}{ccc} i_{1} & i_{2} & u_{c} \end{array}\right]}^{T}$ represents the state, $w={\left[\begin{array}{cc} u_{G} & i_{ref} \end{array}\right]}^{T}$ represents the external input, and the control input is represented by *u*. It can be noted that the quantities *u*, *u_G_*, and *i_ref_* are three-dimensional vectors consisting of the components for each phase *a*, *b*, and *c*. 

The rest of the matrices are expressed in the following expressions [13,16,17].
(3)$$A=\left[\begin{array}{ccc} -\frac{R_{f}+R_{d}}{L_{f}} & \frac{R_{d}}{L_{f}} & -\frac{1}{L_{f}}\\ \frac{R_{d}}{L_{g}} & -\frac{R_{g}+R_{d}}{L_{g}} & \frac{1}{L_{g}}\\ \frac{1}{C_{f}} & -\frac{1}{C_{f}} & 0 \end{array}\right];\;\;B_{1}=\left[\begin{array}{cc} 0 & 0\\ -\frac{1}{L_{g}} & 0\\ 0 & 0 \end{array}\right];\;B_{2}=\left[\begin{array}{c} \frac{1}{L_{f}}\\ 0\\ 0 \end{array}\right];\;C_{1}=\left[\begin{array}{ccc} 0 & -1 & 0 \end{array}\right];\;\;D_{1}=\left[\begin{array}{cc} 0 & 1 \end{array}\right];\;\;D_{2}=0.$$

The following output can be chosen: $y=e=i_{ref}-i_{2}$.

The transfer function usually denoted as *G* is represented as (4). Usually, *G* can be rewritten according to the theory of robust systems as (5).
(4)$$G=\left[\begin{array}{cc} D_{1} & D_{2} \end{array}\right]+C_{1}{\left(sI-A\right)}^{-1}\left[\begin{array}{cc} B_{1} & B_{2} \end{array}\right]$$
(5)$$G=\left[\begin{array}{ccc} A & B_{1} & B_{2}\\ C_{1} & D_{1} & D_{2} \end{array}\right]$$

These can be represented schematically as in Figure 3. The role of the robust control is to find a controller *K(s)* capable of minimizing the *H∞* norm of the transfer function $T_{\tilde{z}\;\tilde{w}}=F_{l}(P,K)$ from the external inputs $\tilde{w}={\left[\begin{array}{cc} v & w \end{array}\right]}^{T}$ to the quality quantities $\tilde{z}={\left[\begin{array}{cc} z_{1} & z_{2} \end{array}\right]}^{T}$. *ξ*, *μ*, and *W(s)* represents the weighting parameters, which will be specified in the robust controller synthesis algorithm.

The equations of the extended system can be written as follows:(6)$$\left[\begin{array}{c} \tilde{z}\\ \tilde{y} \end{array}\right]=P\left[\begin{array}{c} \tilde{w}\\ u \end{array}\right];\;u=K\cdot \tilde{y}$$
where: the extended plant is noted with *P* and *K* is the controller to be designed. The extended plant *P* contains, as in Figure 3, the weighting *ξ* and *μ* and the low-pass filter *W*(*s*). 

Based on these specifications, Equation (6) will be extended in the form of Equations (7) and (8) [13,16,17].
(7)$$\tilde{y}=e+\xi v=\xi v+\left[\begin{array}{ccc} A & B_{1} & B_{2}\\ C_{1} & D_{1} & D_{2} \end{array}\right]\cdot \left[\begin{array}{c} w\\ u \end{array}\right]=\left[\begin{array}{cccc} A & 0 & B_{1} & B_{2}\\ C_{1} & \xi & D_{1} & D_{2} \end{array}\right]\cdot \left[\begin{array}{c} v\\ w\\ u \end{array}\right]$$
(8)$$\left\{ \begin{array}{l} z_{1}=W\left(e+\xi v\right)=\left[\begin{array}{ccccc} A & 0 & 0 & B_{1} & B_{2}\\ B_{\omega }C_{1} & A_{\omega } & B_{\omega }\xi & B_{\omega }D_{1} & B_{\omega }D_{2}\\ 0 & C_{\omega } & 0 & 0 & 0 \end{array}\right]\left[\begin{array}{c} v\\ w\\ u \end{array}\right]\\ z_{2}=\mu \cdot u \end{array}\right.$$

### 2.2. Matlab/Simulink Implementation of the Robust Control for DC-AC Converter

Using the notations in Section 2.1, the extended plant *P* takes the following form [13,16,17]:(9)$$P=\left[\begin{array}{ccccc} A & 0 & 0 & B_{1} & B_{2}\\ B_{\omega }C_{1} & A_{\omega } & B_{\omega }\xi & B_{\omega }D_{1} & B_{\omega }D_{2}\\ 0 & C_{\omega } & 0 & 0 & 0\\ 0 & 0 & 0 & 0 & \mu \\ C_{1} & 0 & \xi & D_{1} & D_{2} \end{array}\right]$$

By using *hinfsyn()* command from Robust Control toolbox of Matlab, the robust controller *K*(*s*) can be obtained [16,17]: (10)$$W=\left[\begin{array}{cc} -2550 & 2550\\ 1 & 0 \end{array}\right]$$

The transfer functions of the low-pass filters on each phase used to filter the voltages in the grid from Figure 1 are chosen by the form expressed in relation (11), and additionally, the following weights can be chosen as: *ξ* = 100 and *μ* = 0.26.
(11)$$F(s)=\frac{0.165\cdot s+33}{0.002\cdot s^{2}1.6\cdot s+300}$$

The synthesized controller, weights and low-pass filters are implemented in a Simulink-type scheme as in Figure 4. The transfer function of the robust controller *K*(*s*) is shown in relation (12).
(12)$$K(s)=\frac{0.098\cdot s^{2}+550\cdot s+3627}{s^{2}+50\cdot s+980}$$

The values of nominal parameters of the DC-AC converter circuit elements are given in Table 1.

### 2.3. Improvement of the Robust Control for DC-AC Converter Using RL-TD3 Agent

A combined control of the DC-AC converter system based on a robust controller and RL-TD3 agent can be proposed to improve the performance of the DC-AC converter control system. Among machine learning-based controls, the most suitable variant for industrial process control is provided by RL [24,25,26,27].

Thus, the main stages of creating, training, validating and using an RL agent are suggestively presented in Figure 5. Also, by analogy with the control of an industrial process, it can be noted that, based on observations collected from the *Environment* (similarly to reading analog/digital inputs from an industrial process), the RL-TD3 agent provides actions (similarly to providing analog/digital outputs to an industrial process) based on the optimization of a reward calculated according to the proposed objectives (similarly to the optimization of an integral criterion in the industrial process control). 

For the improvement of the proposed control system, an RL-TD3 agent algorithm is chosen. After completing the training, testing and validation stages, the RL-TD3 agent will provide correction signals to the robust controller commands to improve the performance of the control system for the DC-AC converter shown in Figure 6.

The details of the Matlab/Simulink implementation of the RL-TD3 agent for the correction of *u_aref_*, *u_bref_*,and *u_cref_* command signals are presented in Figure 7. 

With the values of the circuit elements presented in Table 1, the robust controller and the filters presented in Section 2.2, and for *i_dref_* = 5 A, *i_qref_* = 0 A, *u_dref_* = 310 V, and *u_qref_* = 0 V, Figure 8 shows the reward evolution in training stage for the implemented RL-TD3 algorithm performance. 

The time of the training stage for the implemented RL-TD3 agent for command signals correction of the robust controller is 2 h, 11 min, and 5 s. The sampling time of the RL-TD3 algorithm is 10^−4^ s, and the training stage is of 200 epochs.

In the RL-TD3 agent training stage, it is used an optimization criterion (13) with the usual notations.
(13)$$r_{Robust}=-(5u_{d\_error}^{2}+5u_{q\_error}^{2}+5i_{d\_error}^{2}+5i_{q\_error}^{2}+0.1\sum_{j}{\left(u_{t-1}^{j}\right)}^{2})$$
where: $u_{t-1}^{j}$ includes the actions in the previous step.

## 3. PCH Control of the DC-AC Converter

Similar to the description in Figure 1, Figure 9 shows the block diagram of the control system for the DC-AC converter based on a PCH-type controller. The main components are the follows: DC voltage source; three-phase voltage source inverter (DC-AC converter); LC filter; load; and the control system for DC-AC converter. Usually, the controller is implemented with a PI control law, but in this section, based on the PCH theory, will be presented the synthesis of a PCH controller, which will provide modulation indices for the control of the active control elements in the DC-AC converter. 

### 3.1. Mathematical Description of the PCH Control

If, in the previous section, the description equations of the controlled system are usually linearized to obtain a robust controller, in this section, the PCH theory will be used to obtain a nonlinear controller, which will have superior performance. Thus, Figure 10 shows the schematic single-phase representation of the controlled system.

Based on the PCH theory and *d*-*q* reference frame representation, the synthesis functions of the modulation indices *m_d_* and *m_q_* will be obtained, and then, by means of a PWM block, the switching signals *S*_1_…*S*_6_ will be obtained for the control of the IGBT active elements for the control of the DC-AC converter. 

Starting from the diagram in Figure 10, where the notations are the usual ones in the *d*-*q* reference frame for the modulation indices, angular frequency, currents and voltages, the following equations can be written:(14)$$\left\{ \begin{array}{l} L_{f}{\dot{i}}_{d}=m_{d}u_{dc}-R_{f}i_{d}-\omega_{dq}L_{f}i_{q}-e_{d}\\ L_{f}{\dot{i}}_{q}=m_{q}u_{dc}-R_{f}i_{q}+\omega_{dq}L_{f}i_{d}-e_{q}\\ C_{f}{\dot{e}}_{d}=i_{d}-\frac{e_{q}}{\sqrt{R_{d}^{2}+\frac{1}{\omega_{dq}^{2}C_{f}^{2}}}}-i_{Ld}\\ C_{f}{\dot{e}}_{q}=i_{q}+\frac{e_{d}}{\sqrt{R_{d}^{2}+\frac{1}{\omega_{dq}^{2}C_{f}^{2}}}}-i_{Lq} \end{array}\right.$$

System (14) can be written as Port Hamiltonian model as follows:(15)$$\dot{x}=\left[J-R)\right]\frac{\partial H(x)}{\partial x}+gu+\zeta$$
where: the state vector is noted with *x*, the interconnection matrix and damping matrix are noted with *J* and *R*, the energy stored by the system is noted with *H*(*x*), the input matrix is noted with *g*, the control input vector is noted with *u*, and the external input is noted with *ζ*. 

Thus, the Port Hamiltonian model of the DC-AC converter can be obtained as [22,23]:(16)$$\left[\begin{array}{c} L_{f}{\dot{i}}_{d}\\ L_{f}{\dot{i}}_{q}\\ C_{f}{\dot{e}}_{d}\\ C_{f}{\dot{e}}_{q} \end{array}\right]=\left[\begin{array}{cccc} -R_{f} & -\omega_{dq}L_{f} & -1 & 0\\ \omega_{dq}L_{f} & -R_{f} & 0 & -1\\ 1 & 0 & 0 & -\frac{1}{\sqrt{R_{d}^{2}+\frac{1}{\omega_{dq}^{2}C_{f}^{2}}}}\\ 0 & 1 & \frac{1}{\sqrt{R_{d}^{2}+\frac{1}{\omega_{dq}^{2}C_{f}^{2}}}} & 0 \end{array}\right]\left[\begin{array}{c} i_{d}\\ i_{q}\\ e_{d}\\ e_{q} \end{array}\right]+\left[\begin{array}{cc} u_{dc} & 0\\ 0 & u_{dc}\\ 0 & 0\\ 0 & 0 \end{array}\right]\left[\begin{array}{c} m_{d}\\ m_{q} \end{array}\right]+\left[\begin{array}{c} 0\\ 0\\ -i_{Ld}\\ -i_{Lq} \end{array}\right]$$
where: the matrices from Equation (15) are expressed in the following relations:(17)$$J=\left[\begin{array}{cccc} 0 & -\omega_{dq}L_{f} & -1 & 0\\ \omega_{dq}L_{f} & 0 & 0 & -1\\ 1 & 0 & 0 & -\frac{1}{\sqrt{R_{d}^{2}+\frac{1}{\omega_{dq}^{2}C_{f}^{2}}}}\\ 0 & 1 & \frac{1}{\sqrt{R_{d}^{2}+\frac{1}{\omega_{dq}^{2}C_{f}^{2}}}} & 0 \end{array}\right];\;R(x)=\left[\begin{array}{cccc} R_{f} & 0 & 0 & 0\\ 0 & R_{f} & 0 & 0\\ 0 & 0 & 0 & 0\\ 0 & 0 & 0 & 0 \end{array}\right]$$
where: $J=-J^{T}$ and $R=R^{T}\geq 0$.

Denoting the energy stored in the elements *L_f_* and *C_f_* as *H*(*x*), the following relation can be written:(18)$$H(x)=\frac{1}{2}\left(L_{f}i_{d}^{2}+L_{f}i_{q}^{2}+C_{f}e_{d}^{2}+C_{f}e_{q}^{2}\right)$$

An admissible state vector is defined based on passivity from control theory [22,23]:(19)$$x_{ref}={\left[\begin{array}{cccc} L_{f}i_{dref} & L_{f}i_{qref} & C_{f}e_{dref} & C_{f}e_{qref} \end{array}\right]}^{T}$$

Based on these, equations expressed in (15) becomes on the form:(20)$${\dot{x}}_{ref}=\left[J-R)\right]\frac{\partial H\left(x_{ref}\right)}{\partial x_{ref}}+gu^{*}+\zeta$$
where: *u^*^* is bounded.

By denoting the variable quantities: $\tilde{x}=x-x_{ref}$ and $\tilde{u}=u-u^{*}$, the system (20) becomes:(21)$$\dot{\tilde{x}}+{\dot{x}}_{ref}=\left[J-R)\right]\frac{\partial H\left(\tilde{x}+x_{ref}\right)}{\partial \left(\tilde{x}+x_{ref}\right)}+g\left(\tilde{u}+u^{*}\right)+\zeta$$

By denoting the gradient of the energy function as $\frac{\partial H\left(x\right)}{\partial x}=P^{-1}x$, equation expressed in (18) can be rewritten in the next form:(22)$$H\left(x\right)=H\left(\tilde{x}+x_{ref}\right)=\frac{1}{2}x^{T}P^{-1}x=\frac{1}{2}{\left(\tilde{x}+x_{ref}\right)}^{T}P^{-1}\left(\tilde{x}+x_{ref}\right)$$
where the gradient of the variable energy function can be expressed in the next form:(23)$$\frac{\partial H\left(\tilde{x}+x_{ref}\right)}{\partial \left(\tilde{x}+x_{ref}\right)}=P^{-1}\left(\tilde{x}+x_{ref}\right)=P^{-1}\tilde{x}+P^{-1}x_{ref}=\frac{\partial H\left(\tilde{x}\right)}{\partial \tilde{x}}+\frac{\partial H\left(x_{ref}\right)}{\partial x_{ref}}$$

With these the equation given in (21) can be written as follows:(24)$$\dot{\tilde{x}}+{\dot{x}}_{ref}=\left[J-R)\right]\frac{\partial H\left(\tilde{x}\right)}{\partial \tilde{x}}+\left[J-R)\right]\frac{\partial H\left(x_{ref}\right)}{\partial x_{ref}}+g\tilde{u}+gu^{*}+\zeta$$

From Equation (24), the dynamic regime can be obtained as follows:(25)$$\dot{\tilde{x}}=\left[J-R)\right]\frac{\partial H\left(\tilde{x}\right)}{\partial \tilde{x}}+g\left(\tilde{u}\right)$$

The output signal of the system can be denoted in the next form:(26)$$\tilde{y}=g^{T}\frac{\partial H\left(\tilde{x}\right)}{\partial \tilde{x}}$$

Using the energy function expressed in (27) by performing a series of calculations, it can be concluded that the system given in (25) is passive, because the inequality $\dot{H}\left(\tilde{x}\right)\leq {\tilde{y}}^{T}\tilde{u}$ is fulfilled [22,23].
(27)$$H\left(\tilde{x}\right)=\frac{1}{2}{\tilde{x}}^{T}P^{-1}x$$

With these, the PCH controller has the next form:(28)$$\begin{array}{l} \dot{z}=-\tilde{y}\\ \tilde{u}=-K_{P}\tilde{y}+K_{I}z \end{array}$$

This form is the analogue of a PI controller with constants *k_P_* and *k_I_*, where the output signal is given by the equation expressed in (29) like in the next form:(29)$$\tilde{y}=g^{T}\frac{\partial H\left(\tilde{x}\right)}{\partial \tilde{x}}=\left[\begin{array}{c} u_{dc}\left(i_{d}-i_{dref}\right)\\ u_{dc}\left(i_{q}-i_{qref}\right) \end{array}\right]$$

Based on these, from Equation (24) currents *i_dref_* and *i_qref_* can be obtained as Equation (30) and the modulation indices *m_dref_* and *m_qref_* as Equation (31).
(30)$$\left\{ \begin{array}{l} i_{dref}=\frac{e_{qref}}{\sqrt{R_{d}^{2}+\frac{1}{\omega_{dq}^{2}C_{f}^{2}}}}+i_{Ld}\\ i_{qref}=-\frac{e_{dref}}{\sqrt{R_{d}^{2}+\frac{1}{\omega_{dq}^{2}C_{f}^{2}}}}+i_{Lq} \end{array}\right.$$
(31)$$\left\{ \begin{array}{l} m_{dref}=\frac{1}{u_{dc}}\left(L_{f}i_{dref}+R_{f}i_{dref}+\omega_{dq}L_{f}i_{qref}+e_{dref}\right)\\ m_{qref}=\frac{1}{u_{dc}}\left(L_{f}i_{qref}+R_{f}i_{qref}-\omega_{dq}L_{f}i_{dref}+e_{qref}\right) \end{array}\right.$$

### 3.2. Matlab/Simulink Implementation of the PCH Control Combined with RL-TD3 Agent for Command Signals Correction

Similar to Section 2.2, the main purpose of this section is to present a method for improving the control system for DC-AC converter performance by using an RL-TD3 agent, in which the basic controller is shown to be both the classic PI type and the PCH type controller. 

Based on the classic PI control structure, Figure 11 shows the block diagram structure for the Matlab/Simulink model implementation of the control system for the DC-AC converter based on PI controller and an RL-TD3 agent.

Figure 12 shows the details implementation of the RL-TD3 agent for the correction of *i_dref_* and *i_qref_* signals, which is represented in the Reinforcement Learning subsystem shown in Figure 11. 

With the values of the circuit elements presented in Table 1, the PI controllers and RL-TD3 agent for control of the DC-AC converter, and for *i_dref_* = 5 A, *i_qref_* = 0 A, *u_dref_* = 310 V, and *u_qref_* = 0 V, Figure 13 presents the reward evolution of the RL-TD3 algorithm performance. 

The time of the training stage for the implemented RL-TD3 agent for command signals correction of the PI controller is one hour, 42 min, and 11 s. 

The sampling time of the RL-TD3-type agent algorithm is 0.0001 s, and the training stage is 200 epochs.

The optimization criterion (the reward) used in the training stage of the control system for DC-AC converter based on PI controllers and RL-TD3 agent is presented in Equation (32).
(32)$$r_{PI}=-(5i_{q\_error}^{2}+5i_{d\_error}^{2}+0.1\sum_{j}{\left(u_{t-1}^{j}\right)}^{2})$$

Figure 14 shows the block diagram structure for the Matlab/Simulink model implementation of the control system for the DC-AC converter based on PHC controller and an RL-TD3 agent. It can be noted in the Simulink implementation of Equations (30) and (31) in the structure of the PCH-type controller. 

The detail of the implementation of the RL-TD3 agent for the correction of *e_dref_*, *e_qref_*, *i_dref_*, and *i_qref_* command signals, which is represented in the Reinforcement Learning subsystem shown in Figure 14, is presented in Figure 15.

With the values of the circuit elements presented in Table 1, the PCH-type controller and RL-TD3 agent for control of DC-AC converter, and for *i_dref_* = 5 A, *i_qref_* = 0 A, *u_dref_* = 310 V, and *u_qref_* = 0 V, Figure 16 presents the reward evolution of the RL-TD3 algorithm performance. 

The time of the training stage for the implemented RL-TD3 agent for command signals correction of the PCH-type controller is one hour, 58 min, and 56 s. The sampling time of the RL-TD3-type agent algorithm is 0.0001 s and the training stage is of 200 epochs. 

The optimization criterion (the reward) used in the training stage of the control system for DC-AC converter based on PCH controller and RL-TD3 agent is presented in Equation (33).
(33)$$r_{PHC}=-(5u_{d\_error}^{2}+5u_{q\_error}^{2}+5i_{d\_error}^{2}+5i_{q\_error}^{2}+0.1\sum_{j}{\left(u_{t-1}^{j}\right)}^{2})$$

The control law for DC-AC converter output is given by the modulation indices *m_d_* and *m_q_*, and by means of an inverse Park transformation (*d*-*q→abc*), the real modulation indices *m_a_*, *m_b_*, and *m_c_* are obtained. These modulation indices provide the input signals for a PWM block whose outputs are represented by the switching signals *S*_1_…*S*_6_, which represent the control elements for the active elements of the DC-AC converter voltage.

## 4. Numerical Simulations

Starting from Figure 1, Figure 2, Figure 9 and Figure 10, which show the block diagram for the control system of the DC-AC converter using a robust controller and PCH-type controller, respectively, Figure 17 summarizes the Matlab/Simulink implementation of the proposed control system of the DC-AC converter based on PI, Robust or PCH type controllers and RL-TD3 agents for command signals correction. The numerical values of the circuit elements are given in Table 1 in Section 2, and the quality quantities *u_d_* and *u_q_* voltages defined *d*-*q* frame, aimed at DC-AC converter control, will be kept at constant values *u_d_* = 310 V and *u_q_* = 0 V. 

The controllers used are the classic PI controller, the robust controller and the nonlinear PCH controller. Each of these three controllers will be backed up with an RL-TD3 agent trained accordingly in order to improve the performance of each control system. The aimed performances of the DC-AC converter control systems are the steady-state error, the error ripple, and the THD current. In order to reveal aspects of the actual operation, for each of the controllers presented above and the targeted performance, the load used in the simulation will be of three types: balanced, unbalanced, and nonlinear. In the case of the balanced load, the resistance on each phase is 5 Ω. In the case of the unbalanced load, the resistance on phase *b* is chosen of a very high value compared to the other two phases, *a* and *c,* with a resistance of 5 Ω. In the case of nonlinear load, the resistances on each phase are the same but are described by voltage-current pairs *u*(*k*) and *i*(*k*), where the discretization variable *k* covers the simulation period. 

Figure 18, Figure 19, Figure 20, Figure 21, Figure 22, Figure 23, Figure 24, Figure 25, Figure 26, Figure 27, Figure 28, Figure 29, Figure 30, Figure 31, Figure 32, Figure 33, Figure 34 and Figure 35 present the time evolution of *u_d_* and *u_q_* voltages for DC-AC converter control system based on PI controller, robust controller, PCH-type controller with or without RL-TD3 agent, and the load is balanced, unbalanced or nonlinear. 

Thus, Figure 18, Figure 19 and Figure 20 show the time evolution of *u_d_* and *u_q_* voltages for the DC-AC converter control system based on the PI controller in the case when the load is balanced, unbalanced or nonlinear. Figure 21, Figure 22 and Figure 23, for the same types of load variation, show the time evolution of *u_d_* and *u_q_* voltages for the DC-AC converter control system based on the PI controller improved by using an RL-TD3 agent. Substantial improvement in control system performance can be observed when using PI control in combination with an RL-TD3 agent.

Figure 24, Figure 25 and Figure 26 show the time evolution of *u_d_* and *u_q_* voltages for the DC-AC converter control system based on a robust controller when the load is balanced, unbalanced or nonlinear. Figure 27, Figure 28 and Figure 29 for the same types of load variation, show the time evolution of *u_d_* and *u_q_* voltages for the DC-AC converter control system based on robust controller improved by using an RL-TD3 agent. Substantial improvement in control system performance can be observed when using the robust control in combination with an RL-TD3 agent.

Figure 30, Figure 31 and Figure 32 show the time evolution of *u_d_* and *u_q_* voltages for the DC-AC converter control system based on the PCH-type controller in the case when the load is balanced, unbalanced or nonlinear. 

Figure 33, Figure 34 and Figure 35, for the same types of load variation, show the time evolution of *u_d_* and *u_q_* voltages for the DC-AC converter control system based on the PCH-type controller improved by using an RL-TD3 agent. Substantial improvement in control system performance can be observed when using PCH-type control in combination with an RL-TD3 agent.

In Table 2, in terms of the steady-state error, it can be noted that the performance of each control system based on the main PI controller, robust controller, and PCH-type controller is improved when using a properly trained RL-TD3 agent. Moreover, in the hierarchy of the three basic controllers, the robust-type controller has better performance than the classic PI-type controller, but obviously, the system controlled with a nonlinear PCH-type controller has superior performance. 

It can also be noted that the steady-state error in robust and PCH controllers, with or without RL-TD3 agent, is two to five times lower than the steady-state error when using a classic PI controller. It is also worth noting that the use of an RL-TD3 agent in tandem with the robust controller provides superior performance compared to a nonlinear PCH controller without an RL-TD3 agent. 

In general, the analysis in Table 2 shows that the steady-state errors with respect to the basic balanced load regime are 50% higher in the case of nonlinear load and up to five times higher in the case of unbalanced load for each of the controllers used.

Also, another important indicator for characterizing the performance of the DC-AC converter control system is the ripple of the error signal of the *u_d_* voltage, which is calculated according to Equation (34). It can be concluded from the analysis of the results presented in Table 2 that the order of the controllers in terms of the performance of the control system is also maintained for this indicator, similar to the case of the steady-state error performance indicator. 

Thus, the superiority of the PCH nonlinear controller is also concluded for the error signal ripple indicator, and there are also obvious improvements brought by the use of RL-TD3 agent. In Table 2 it can be noted that the error ripple value with respect to the basic case of the balanced load is up to 20% higher in the case of the nonlinear load and about four times higher in the case of the unbalanced load.
(34)$$u_{d\_rip}=\sqrt{\frac{1}{N}\sum_{i=1}^{N}{\left(u_{d}(i)-u_{dref}(i)\right)}^{2}}$$
where: *N* represents the sample number, *u_d_* represents the voltage and *u_dref_* represents the reference voltage.

Another important indicator of the DC-AC converter control system is the THD which is described by the following relation: (35)$$THD(\%)=\left(\sqrt{\sum_{n=1}^{N}I_{N}^{2}}/I_{RMS}\right)$$
where: *I*_N_ is the RMS value of the harmonic *N* and *I_RMS_* is the RMS value of the fundamental of the signal. 

Figure 36, Figure 37, Figure 38, Figure 39, Figure 40 and Figure 41 show the FFT analysis and THD for the current phase *a* of the DC-AC converter controller for the types of controllers and load variations presented above. Figure 36 and Figure 37 show FFT analysis and THD for the current on phase *a* of the DC-AC converter controlled with PI-type controller without/with RL-TD3 agent in the case of balanced, unbalanced or nonlinear type for the load. Figure 38 and Figure 39 show FFT analysis and THD for the current on phase *a* of the DC-AC converter controlled with a robust-type controller without/with RL-TD3 agent in the case of balanced, unbalanced or nonlinear type for the load. Figure 40 and Figure 41 show FFT analysis and THD for the current on phase *a* of the DC-AC converter controlled with PCH-type controller without/with RL-TD3 agent in the case of balanced, unbalanced or nonlinear type for the load.

Since the controlled system is a DC-AC converter, the THD-type indicator of the current signal on phase *a* is a very important indicator, especially as it must be lower than a value required by power quality standards (usually IEC and IEEE type standards [28] recommend a current THD of less than 12% for a number of harmonics *N* = 50).

Table 2 shows the THD values for the currents on phase *a* for all three types of controllers presented with or without RL-TD3 agent for the three types of load presented.

As in the case of the indicators of the steady-state error and the ripple of the error of the *u_d_* voltage, the order of the performance of the controllers is also kept in the case of the indicator of phase-*a* THD current, in the sense of the superiority of the nonlinear PCH controller and the improvement of the performance of each controller when using an RL-TD3 agent. 

It can be noted, however, that due to the way the nonlinear resistance is defined, the phase-*a* THD current values are twice as high in the unbalanced load case and up to three times as high in the nonlinear load case compared to the main balanced load case.

## 5. Conclusions

This article presented the performance of the control system of a DC-AC converter. The article considers the main elements by which a microgrid represented by a DC power source is connected to the main grid. The main element is a voltage source inverter which is represented by a DC-AC converter whose IGBT active elements are controlled by robust linear or nonlinear PCH controllers. The outputs of these controllers are the modulation indices *m_d_* and *m_q_* in the *d*-*q* reference frame, which, by an inverse Park transformation, are transformed into the actual modulation indices *m_a_*, *m_b_*, and *m_c_*, which provide the switching signals *S*_1_…*S*_6_ for the active elements of the DC-AC converter when they pass through a PWM system. The purpose of the DC-AC converter control system is to maintain the reference values *u_d_* = 310 V and *u_q_* = 0 V of *u_d_* and *u_q_* voltages under load variation. The article presents the block structures of the overall microgrid-to-grid connection system, and the three-phase load is assumed to be balanced/unbalanced or nonlinear. The controllers are classic PI, robust or nonlinear PCH type, and their performance is improved by means of a properly trained RL-TD3 agent. The performance of DC-AC converter control systems is compared using such performance indices as the steady-state and ripple of the error of the *u_d_* voltage and phase-*a* THD current of the microgrid-to-main-grid connection system using a DC-AC converter. The numerical simulations are performed in Matlab/Simulink and reveal the superiority of the performance of the nonlinear PCH controller but also the improvement of the performance of each controller presented by using an RL-TD3 agent, which provides correction signals for the control signals of the corresponding controllers when it is properly trained, to improve the performance of the control systems. In future papers, the software used in the numerical simulations will be implemented in real-time, allowing the transition from the Software-in-the-Loop stage to the Hardware-in-the-Loop stage using dedicated platforms such as SpeedGoat or RT-Opal.

## Figures and Tables

**Figure 1 sensors-22-09535-f001:**
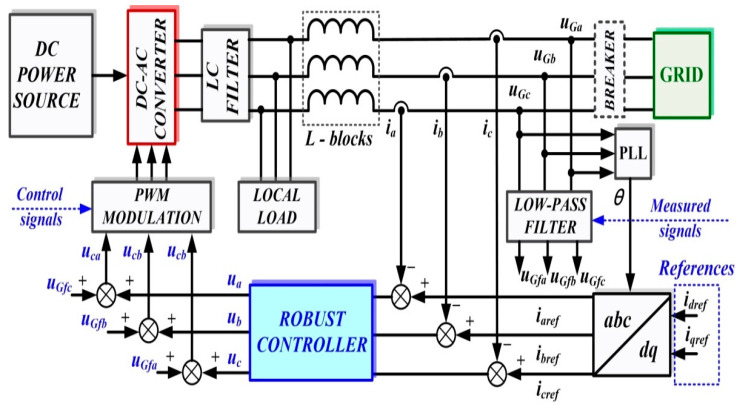
Block diagram for the control system of DC-AC converter using a robust controller.

**Figure 2 sensors-22-09535-f002:**
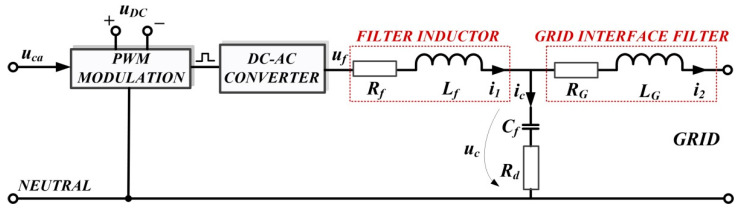
Schematic single-phase representation of plant *G*.

**Figure 3 sensors-22-09535-f003:**
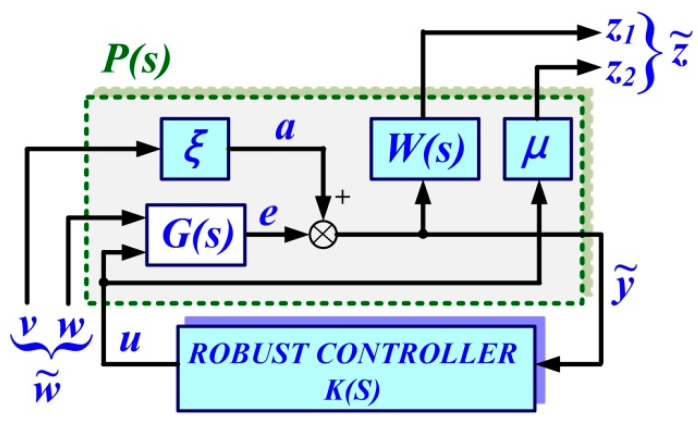
Schematic diagram for the augmented system.

**Figure 4 sensors-22-09535-f004:**
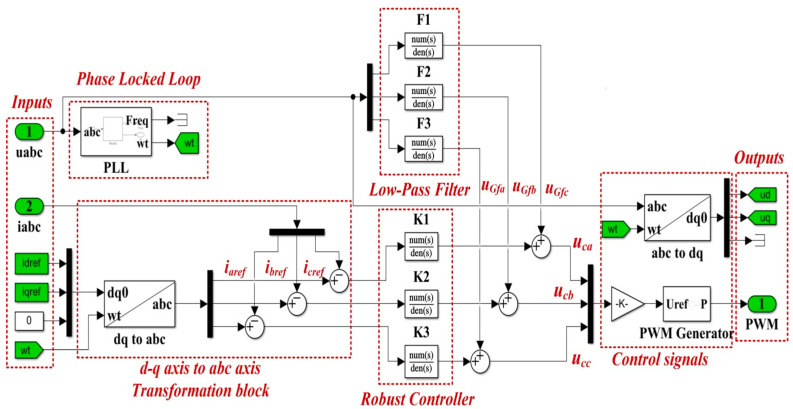
Simulink implementation of the DC-AC converter control system based on robust controller.

**Figure 5 sensors-22-09535-f005:**
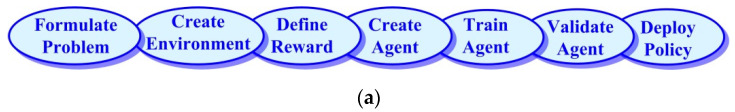
Reinforcement Learning for process control: (**a**) State flow for the RL implementation; (**b**) Block diagram of the RL algorithm scenario.

**Figure 6 sensors-22-09535-f006:**
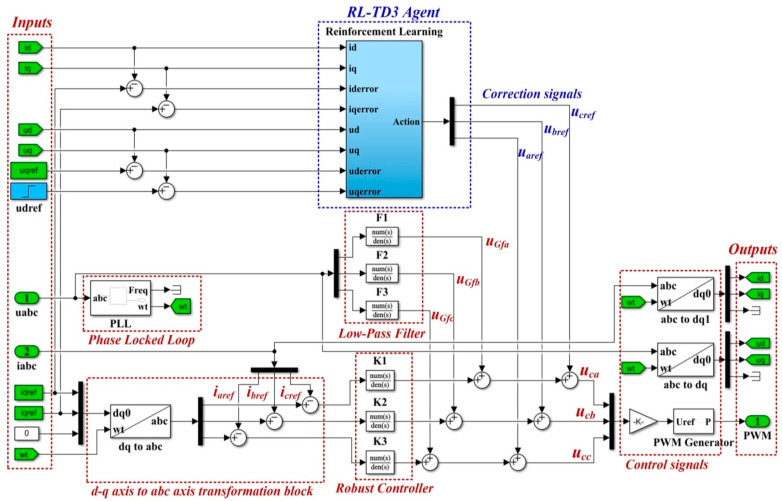
Simulink implementation of the control system for DC-AC converter based on robust controller and RL-TD3 agent.

**Figure 7 sensors-22-09535-f007:**
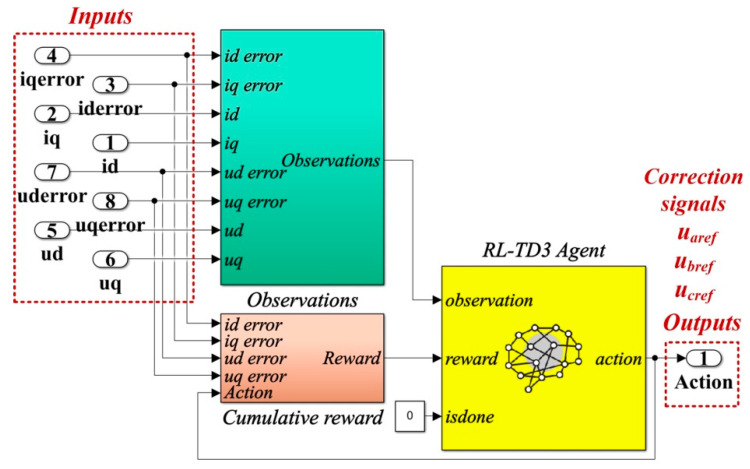
Matlab/Simulink implementation of the RL-TD3 agent for robust controller command signals correction.

**Figure 8 sensors-22-09535-f008:**
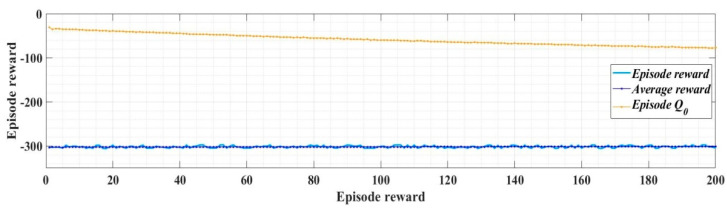
The reward evolution in training stage of the RL-TD3 agent for robust controller command signals correction.

**Figure 9 sensors-22-09535-f009:**
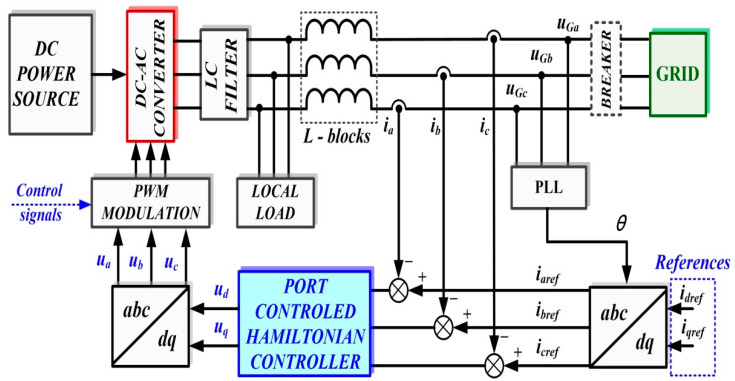
Block diagram of the control system for DC-AC converter based on PCH-type controller.

**Figure 10 sensors-22-09535-f010:**
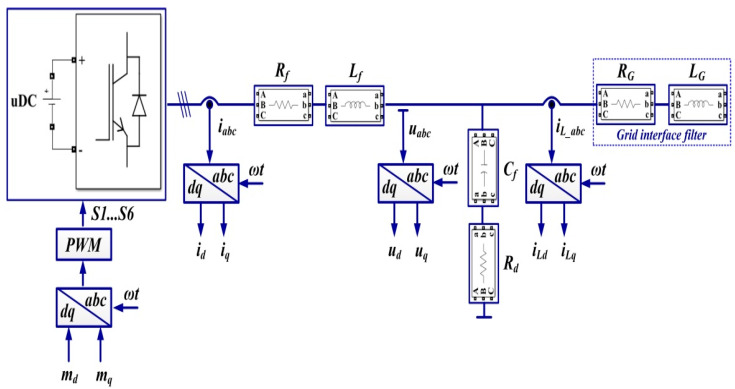
Schematic single-phase representation of the controlled system.

**Figure 11 sensors-22-09535-f011:**
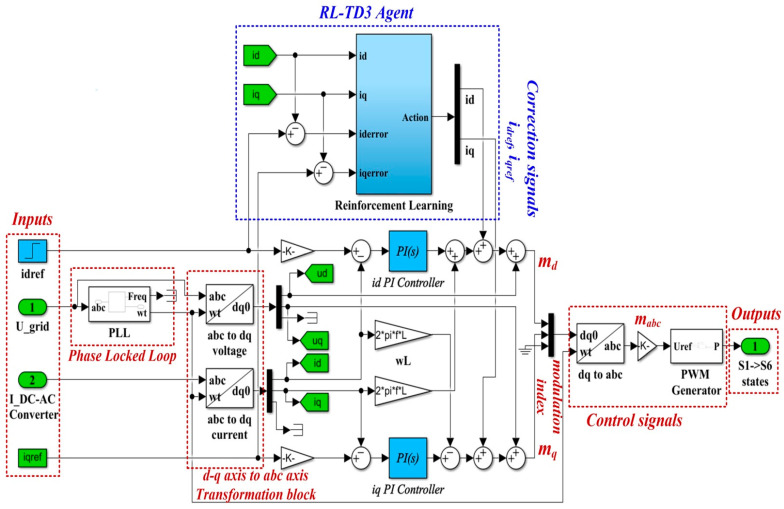
Matlab/Simulink implementation of the control system for DC-AC converter based on PI controllers and RL-TD3 agent.

**Figure 12 sensors-22-09535-f012:**
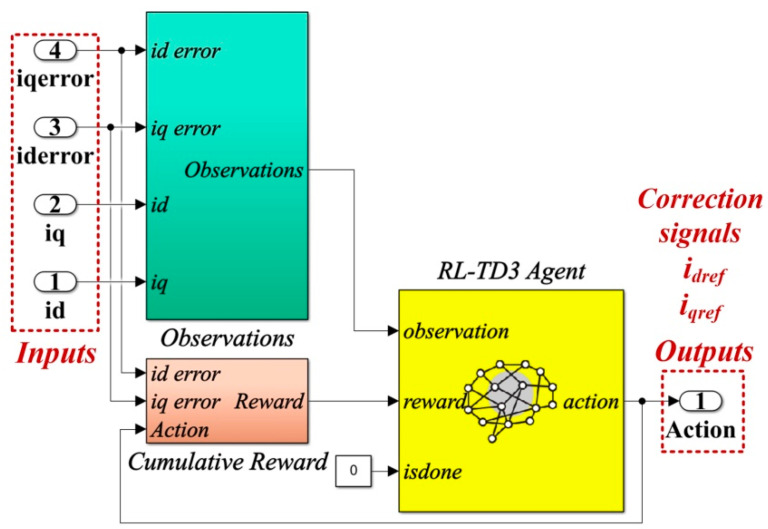
Matlab/Simulink implementation of the RL-TD3 agent for PI controller command signals correction.

**Figure 13 sensors-22-09535-f013:**
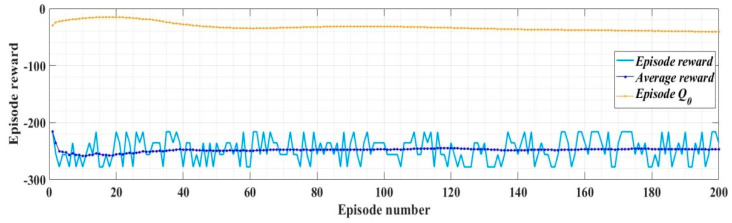
The reward evolution in training stage of the RL-TD3 agent for PI controllers command signals correction.

**Figure 14 sensors-22-09535-f014:**
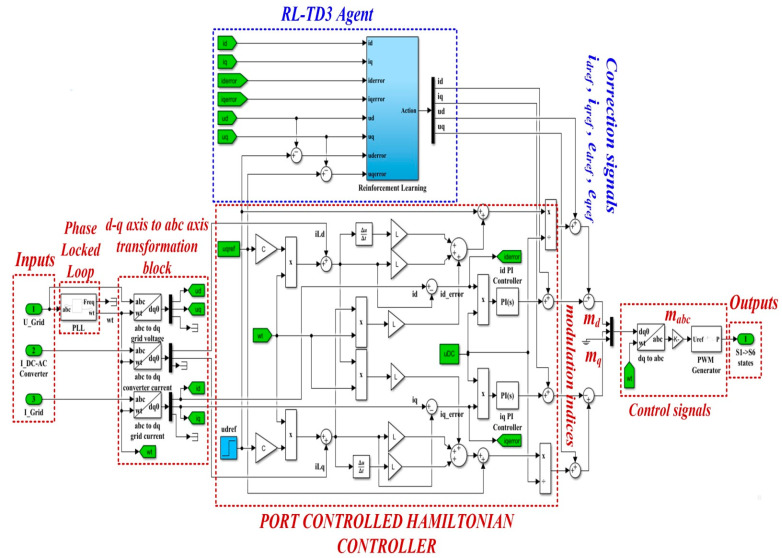
Block diagram structure for the Matlab/Simulink model implementation of the control system for DC-AC converter based on PCH-type controller and a RL-TD3 agent.

**Figure 15 sensors-22-09535-f015:**
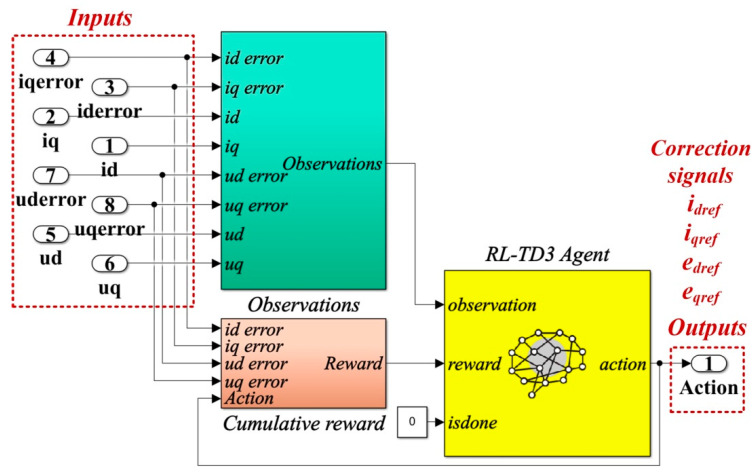
Matlab/Simulink implementation of the RL-TD3 agent for PCH-type controller command signals correction.

**Figure 16 sensors-22-09535-f016:**
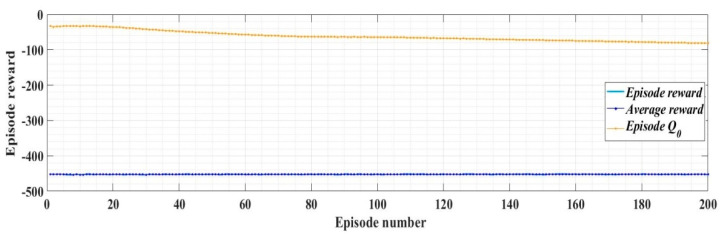
The reward evolution in training stage of the RL-TD3 agent for PCH-type controller command signals correction.

**Figure 17 sensors-22-09535-f017:**
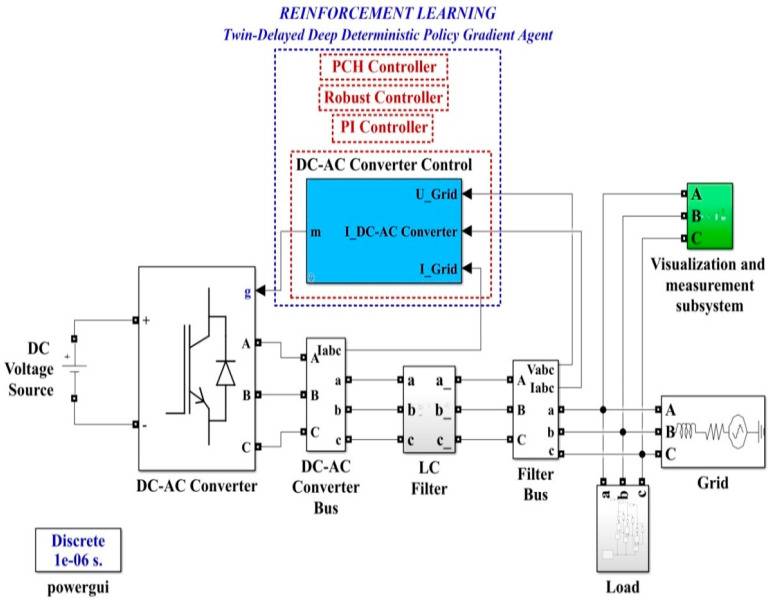
Matlab/Simulink implementation of the proposed control system of the DC-AC converter based on PI, Robust or PCH types controllers and RL-TD3 agents for command signals correction.

**Figure 18 sensors-22-09535-f018:**
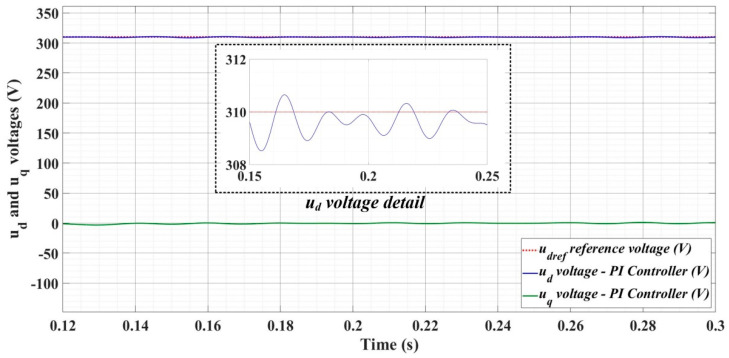
Time evolution of *u_d_* and *u_q_* voltages for DC-AC converter control system based on PI controller in case of balanced resistances for load.

**Figure 19 sensors-22-09535-f019:**
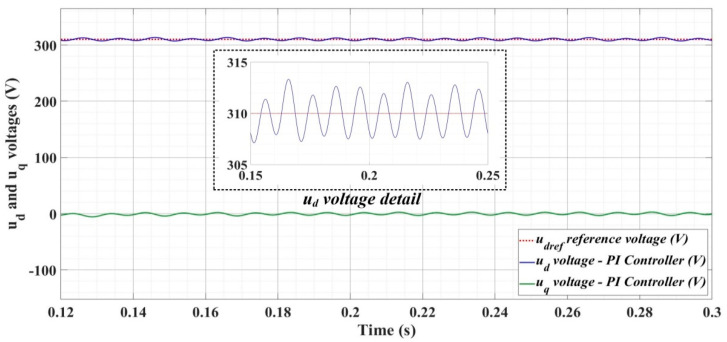
Time evolution of *u_d_* and *u_q_* voltages for DC-AC converter control system based on PI controller in case of unbalanced resistances for load.

**Figure 20 sensors-22-09535-f020:**
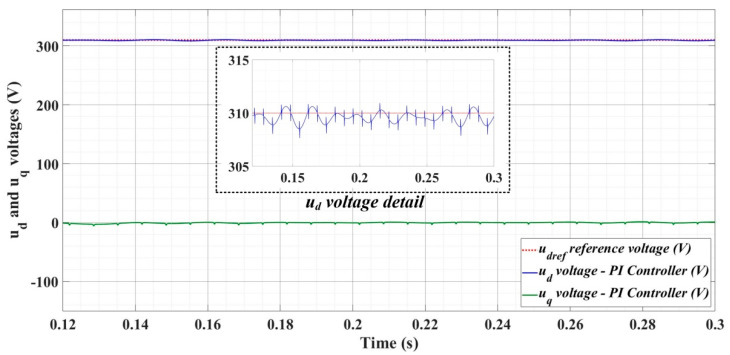
Time evolution of *u_d_* and *u_q_* voltages for DC-AC converter control system based on PI controller in case of nonlinear resistances for load.

**Figure 21 sensors-22-09535-f021:**
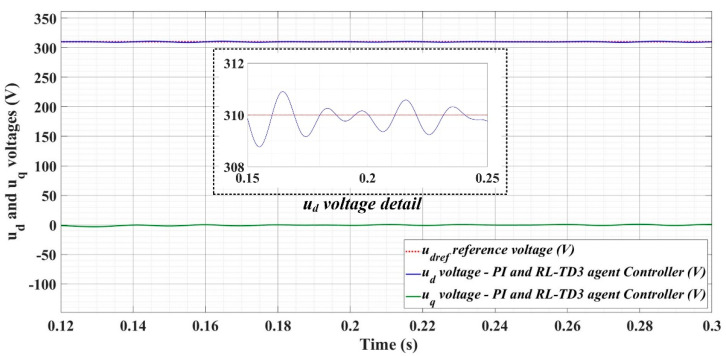
Time evolution of *u_d_* and *u_q_* voltages for DC-AC converter control system based on PI controller using RL-TD3 agent in case of balanced resistances for load.

**Figure 22 sensors-22-09535-f022:**
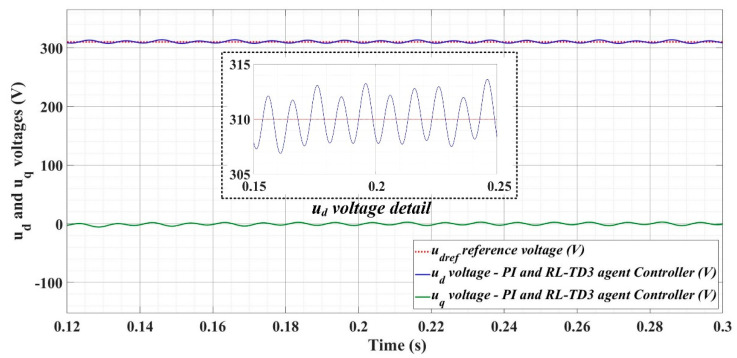
Time evolution of *u_d_* and *u_q_* voltages for DC-AC converter control system based on PI controller using RL-TD3 agent in case of unbalanced resistances for load.

**Figure 23 sensors-22-09535-f023:**
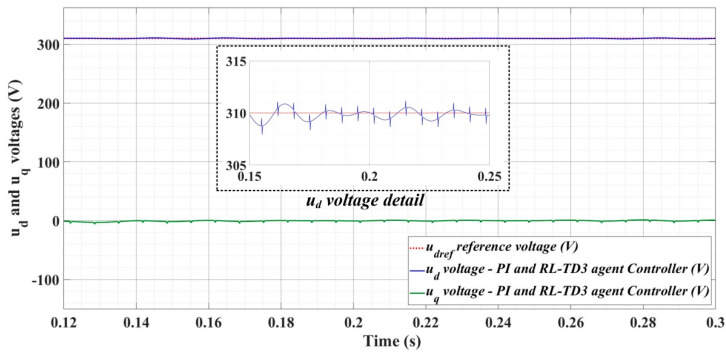
Time evolution of *u_d_* and *u_q_* voltages for DC-AC converter control system based on PI controller using RL-TD3 agent in case of nonlinear resistances for load.

**Figure 24 sensors-22-09535-f024:**
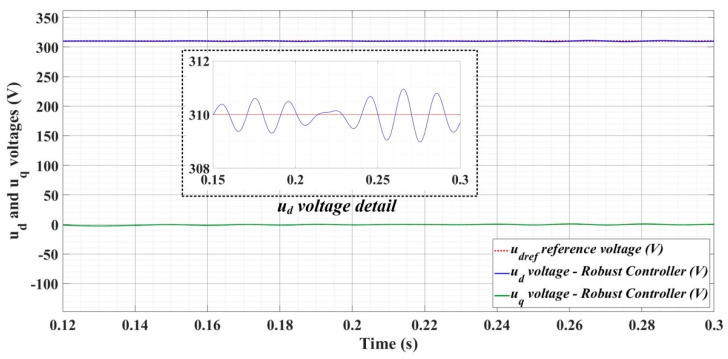
Time evolution of *u_d_* and *u_q_* voltages for DC-AC converter control system based on robust controller in case of balanced resistances for load.

**Figure 25 sensors-22-09535-f025:**
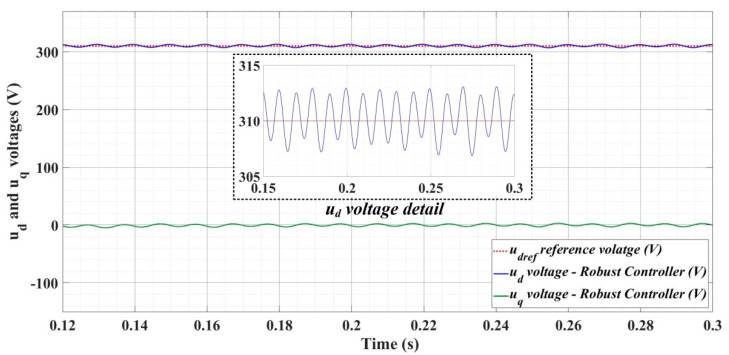
Time evolution of *u_d_* and *u_q_* voltages for DC-AC converter control system based on robust controller in case of unbalanced resistances for load.

**Figure 26 sensors-22-09535-f026:**
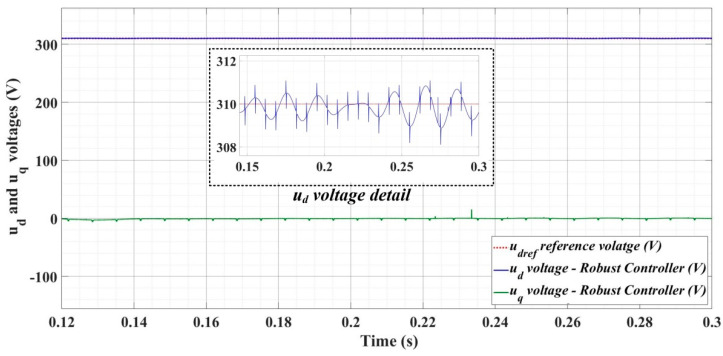
Time evolution of *u_d_* and *u_q_* voltages for DC-AC converter control system based on robust controller in case of nonlinear resistances for load.

**Figure 27 sensors-22-09535-f027:**
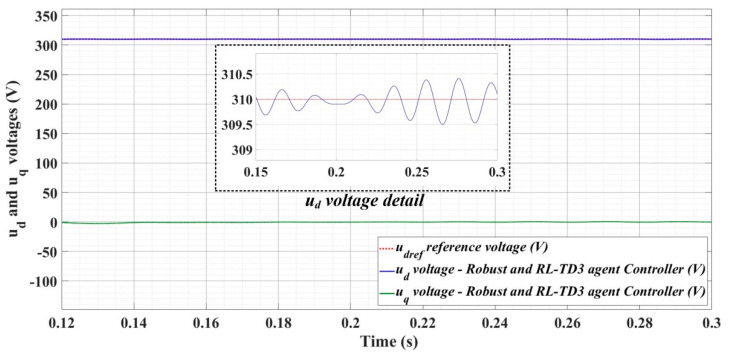
Time evolution of *u_d_* and *u_q_* voltages for DC-AC converter control system based on robust controller using RL-TD3 agent in case of balanced resistances for load.

**Figure 28 sensors-22-09535-f028:**
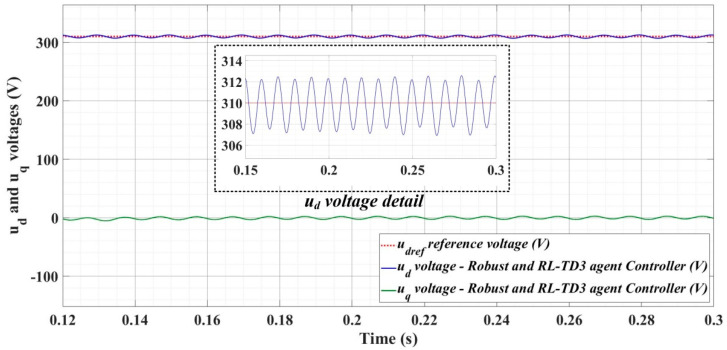
Time evolution of *u_d_* and *u_q_* voltages for DC-AC converter control system based on robust controller using RL-TD3 agent in case of unbalanced resistances for load.

**Figure 29 sensors-22-09535-f029:**
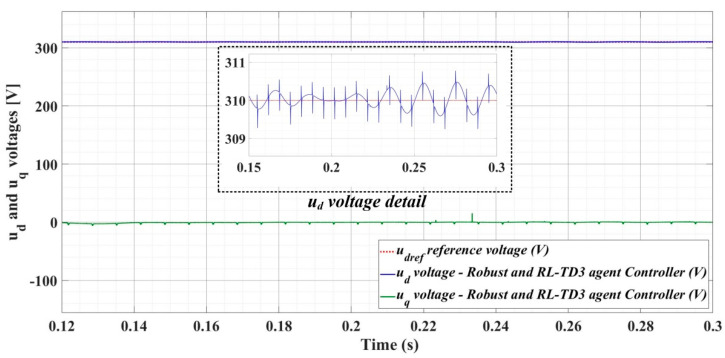
Time evolution of *u_d_* and *u_q_* voltages for DC-AC converter control system based on robust controller using RL-TD3 agent in case of nonlinear resistances for load.

**Figure 30 sensors-22-09535-f030:**
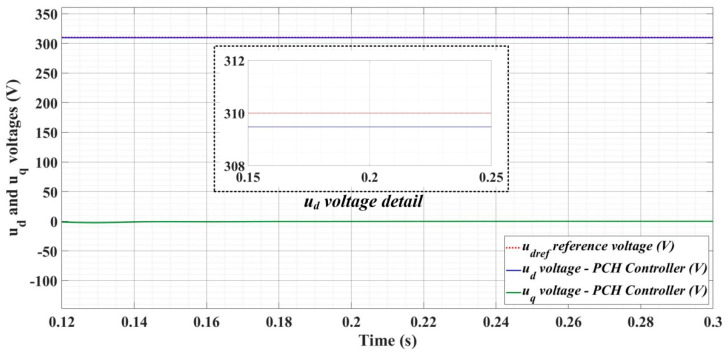
Time evolution of *u_d_* and *u_q_* voltages for DC-AC converter control system based on PCH controller in case of balanced resistances for load.

**Figure 31 sensors-22-09535-f031:**
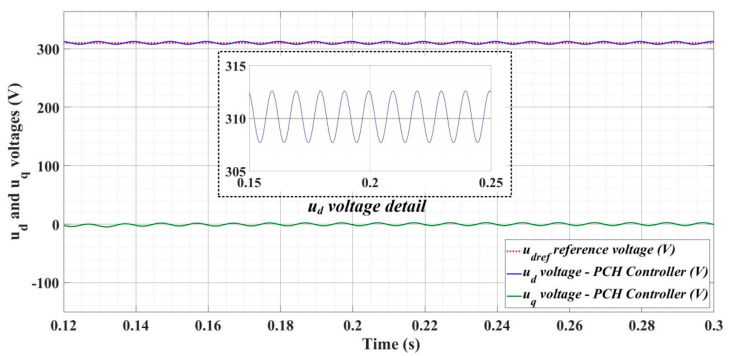
Time evolution of *u_d_* and *u_q_* voltages for DC-AC converter control system based on PCH controller in case of unbalanced resistances for load.

**Figure 32 sensors-22-09535-f032:**
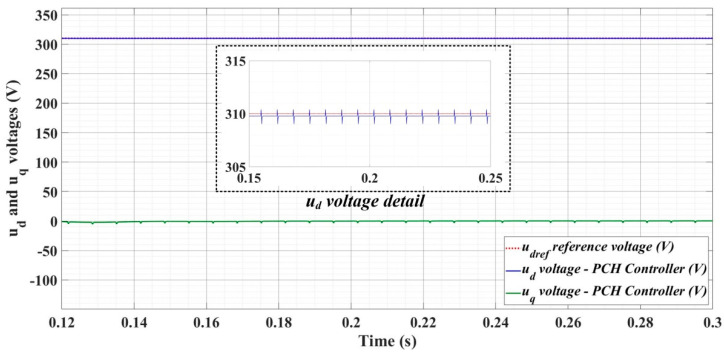
Time evolution of *u_d_* and *u_q_* voltages for DC-AC converter control system based on PCH controller in case of nonlinear resistances for load.

**Figure 33 sensors-22-09535-f033:**
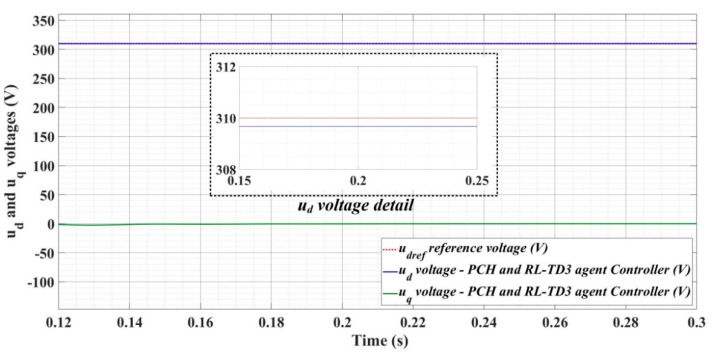
Time evolution of *u_d_* and *u_q_* voltages for DC-AC converter control system based on PCH controller using RL-TD3 agent in case of balanced resistances for load.

**Figure 34 sensors-22-09535-f034:**
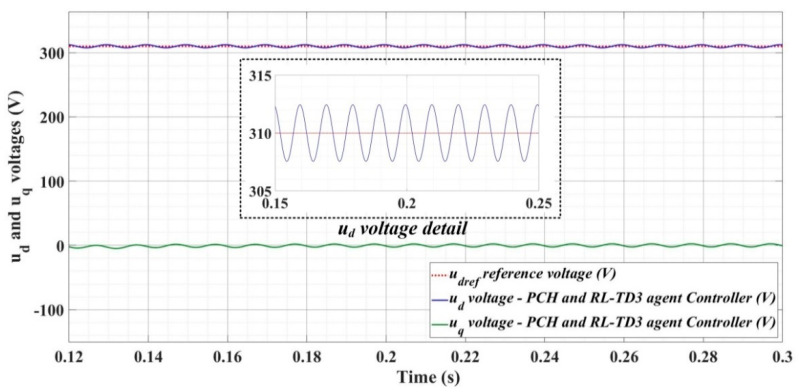
Time evolution of *u_d_* and *u_q_* voltages for DC-AC converter control system based on PCH controller using RL-TD3 agent in case of unbalanced resistances for load.

**Figure 35 sensors-22-09535-f035:**
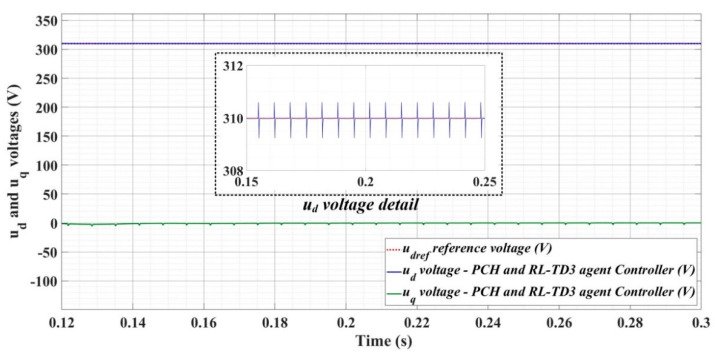
Time evolution of *u_d_* and *u_q_* voltages for DC-AC converter control system based on PCH controller using RL-TD3 agent in case of nonlinear resistances for load.

**Figure 36 sensors-22-09535-f036:**
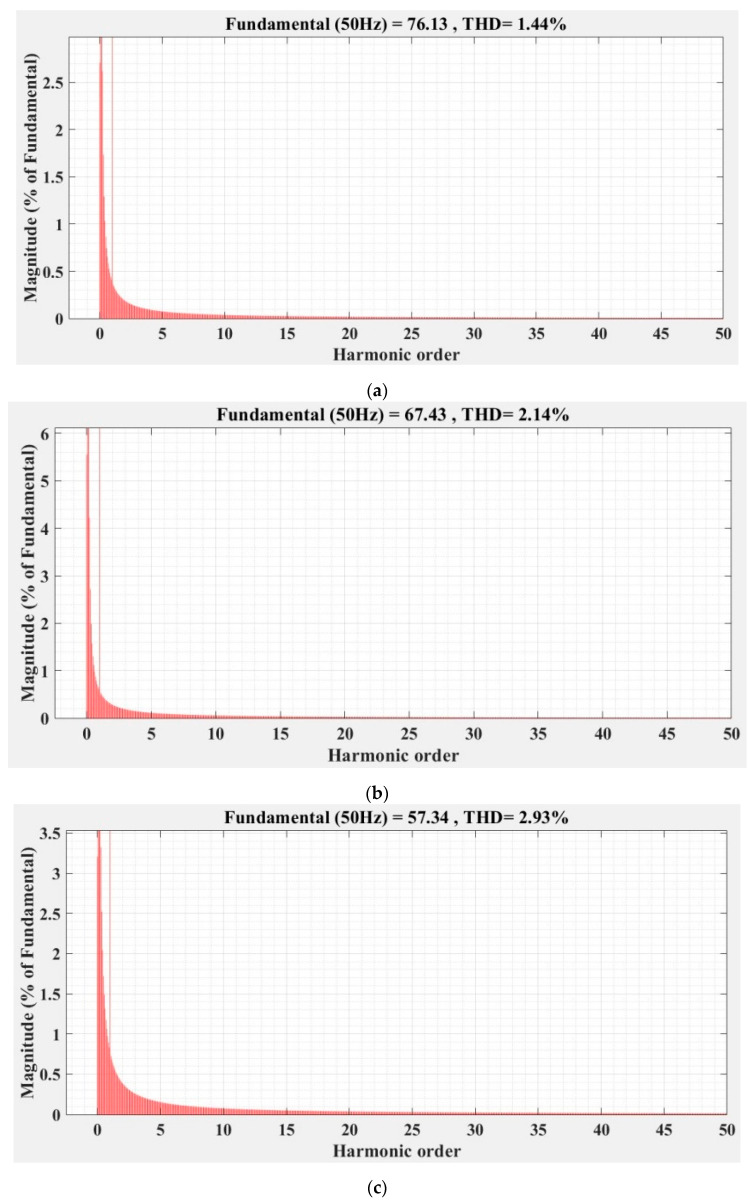
FFT analysis and THD for current phase *a* of the DC-AC converter controlled with PI-type controller: (**a**) balanced resistances for load; (**b**) unbalanced resistances for load; (**c**) nonlinear resistances for load.

**Figure 37 sensors-22-09535-f037:**
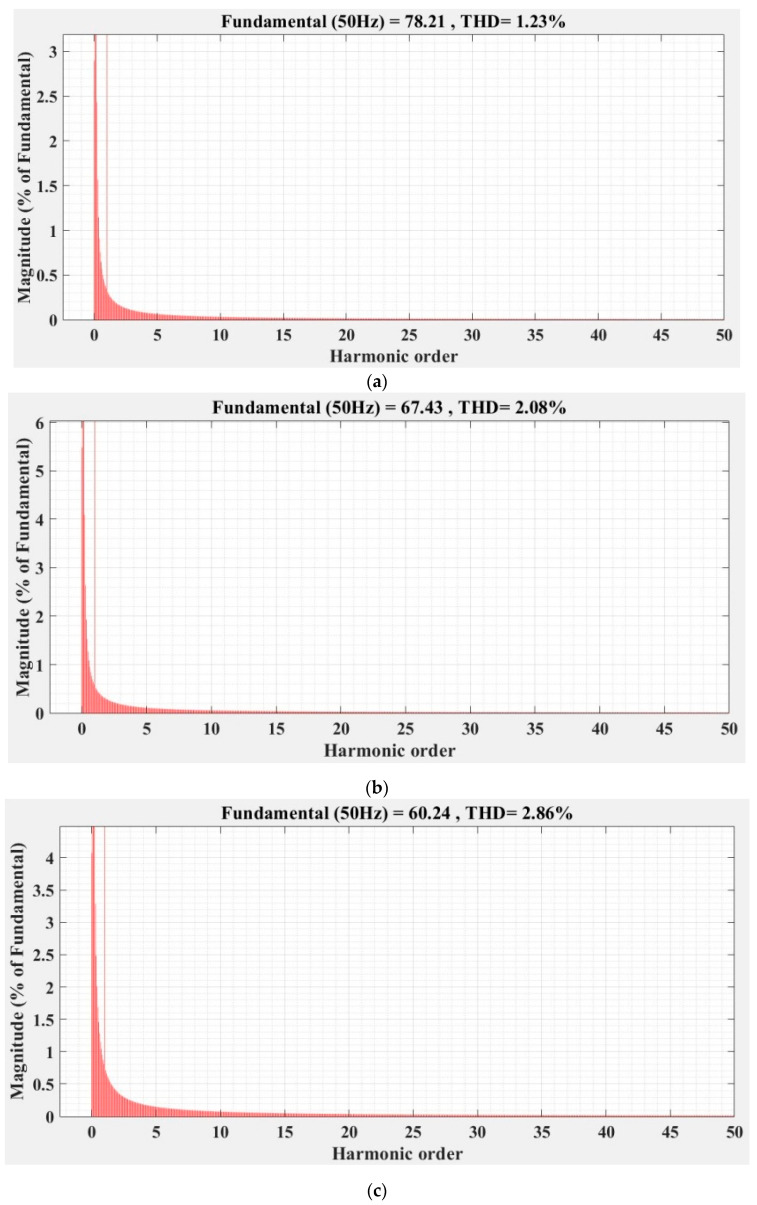
FFT analysis and THD for current phase *a* of the DC-AC converter controlled with PI-type controller using RL-TD3 agent: (**a**) balanced resistances for load; (**b**) unbalanced resistances for load; (**c**) nonlinear resistances for load.

**Figure 38 sensors-22-09535-f038:**
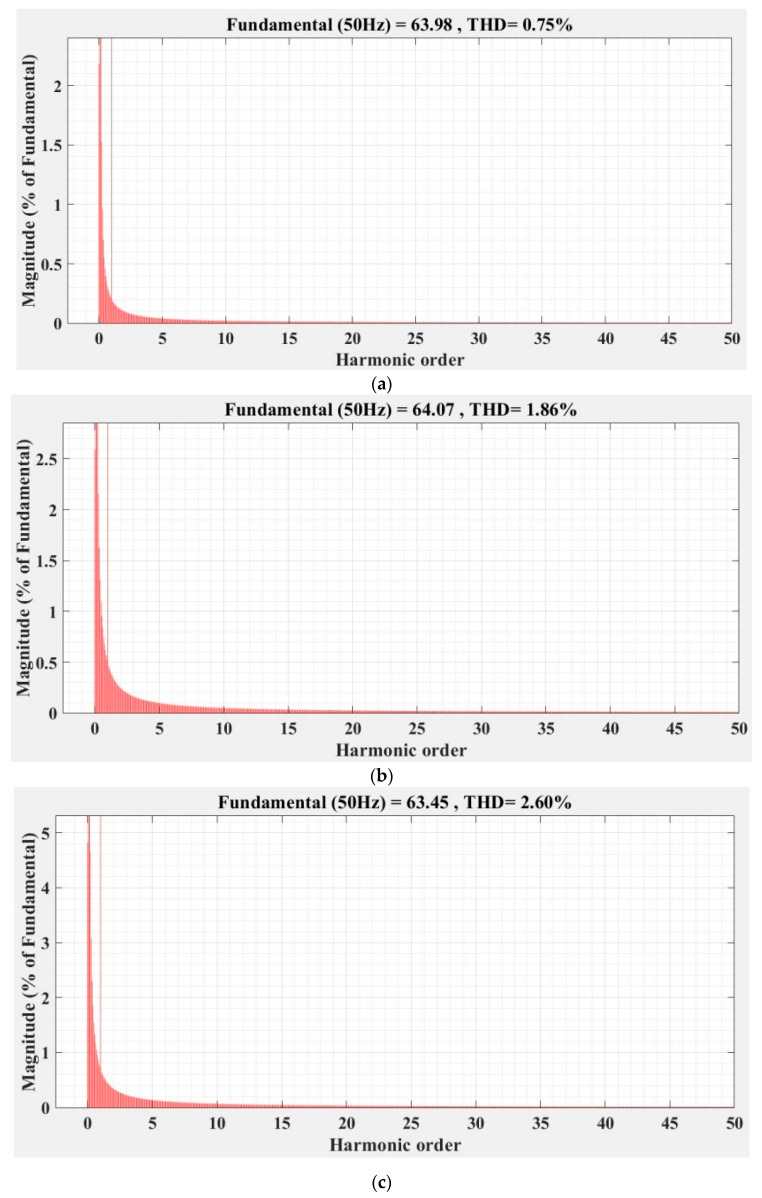
FFT analysis and THD for current phase *a* of the DC-AC converter controlled with robust-type controller: (**a**) balanced resistances for load; (**b**) unbalanced resistances for load; (**c**) nonlinear resistances for load.

**Figure 39 sensors-22-09535-f039:**
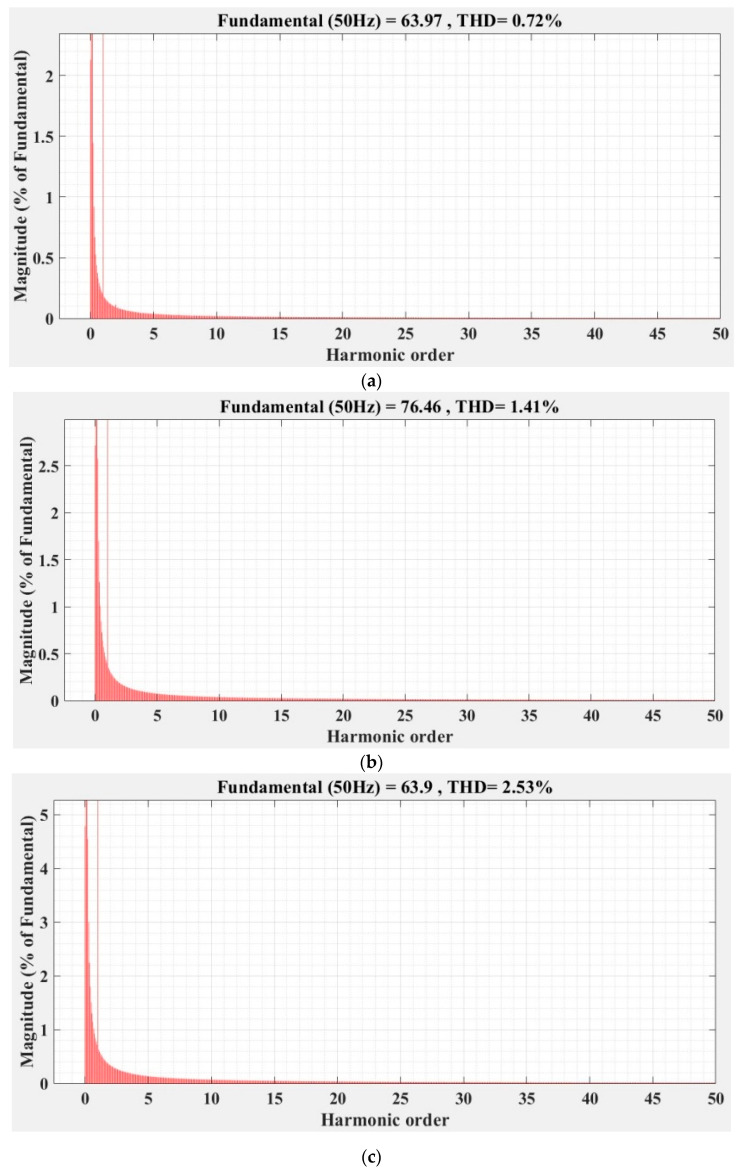
FFT analysis and THD for current phase *a* of the DC-AC converter controlled with robust-type controller using RL-TD3 agent: (**a**) balanced resistances for load; (**b**) unbalanced resistances for load; (**c**) nonlinear resistances for load.

**Figure 40 sensors-22-09535-f040:**
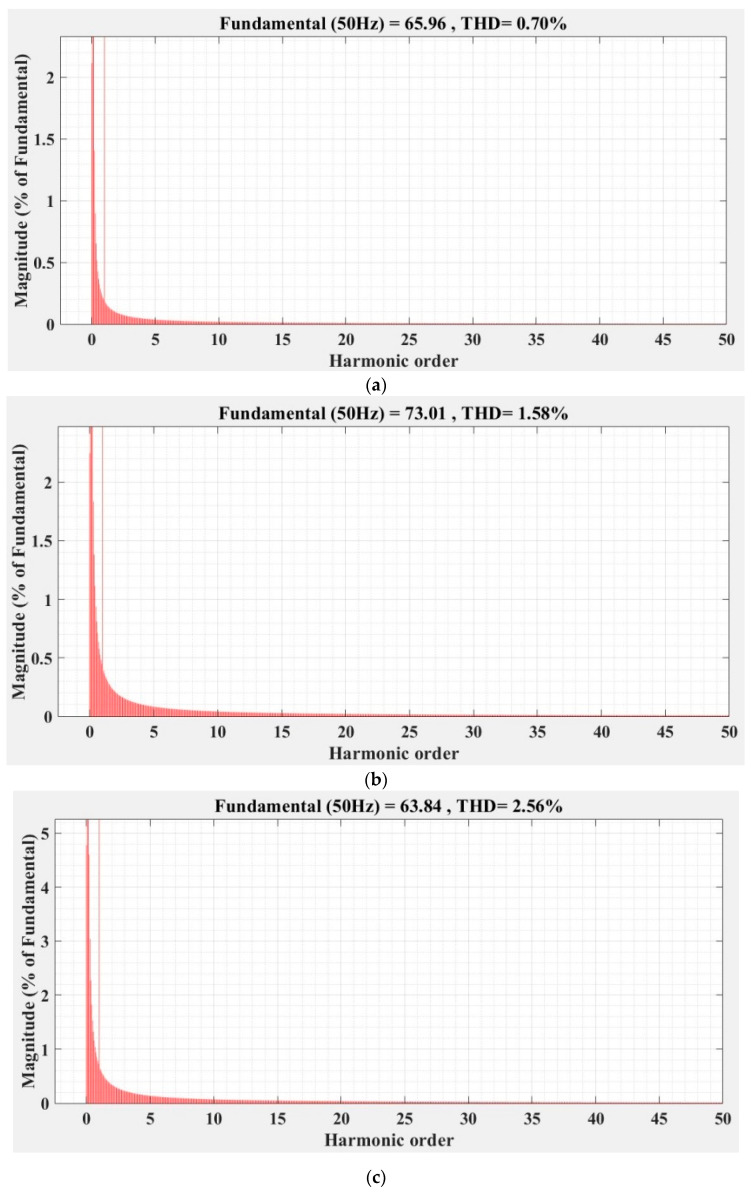
FFT analysis and THD for current phase *a* of the DC-AC converter controlled with PCH-type controller: (**a**) balanced resistances for load; (**b**) unbalanced resistances for load; (**c**) nonlinear resistances for load.

**Figure 41 sensors-22-09535-f041:**
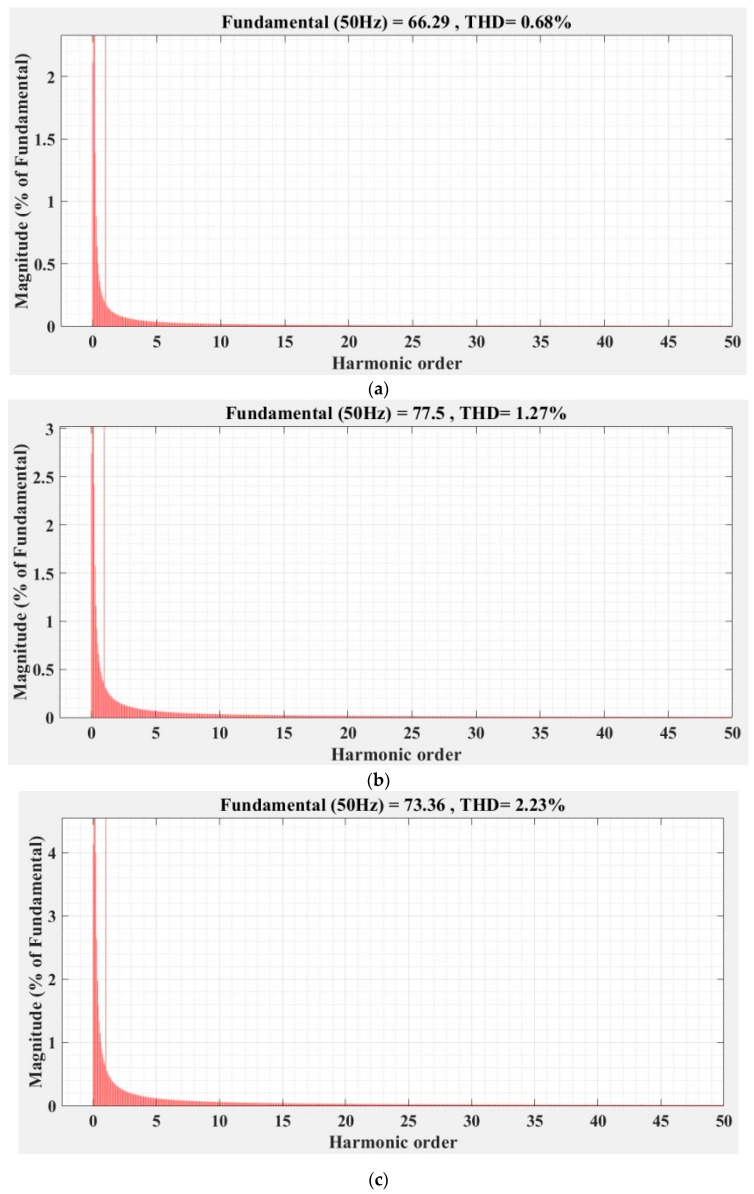
FFT analysis and THD for current phase *a* of the DC-AC converter controlled with PCH-type controller using RL-TD3 agent: (**a**) balanced resistances for load; (**b**) unbalanced resistances for load; (**c**) nonlinear resistances for load.

**Table 1 sensors-22-09535-t001:** DC-AC converter circuit elements—nominal parameters [16,17,22,23].

Parameter	Value	Unit
Filter inductance *L_f_*	150·10^−6^	H
Filter resistance *R_f_*	0.045	Ω
Coupling capacitor *C_f_*	22·10^−6^	F
Grid resistance filter *R_G_*	0.135	Ω
Grid inductance filter *L_G_*	450·10^−6^	H
Resistance of coupling capacitor *R_d_*	1	Ω
Switching frequency of IGBTs	20·10^3^	Hz

**Table 2 sensors-22-09535-t002:** Performance indices of the DC-AC converter control system based on the prosed controllers.

Performance Indices of the DC-AC Converter Control System		PI Controller	PI-RLTD3 Controller	ROBUST Controller	ROBUST-RL-TD3 Controller	PCH Controller	PCH-RL-TD3 Controller
Stationary error [V]	Balanced load	1.64	0.82	0.71	0.41	0.51	0.33
	Unbalanced load	4.19	3.92	3.05	2.43	2.61	2.14
	Nonlinear load	2.29	1.68	1.02	0.84	0.93	0.52
Voltage Ripple [V]	Balanced load	0.622	0.522	0.514	0.332	0.433	0.217
	Unbalanced load	1.774	1.792	1.801	1.738	1.799	1.729
	Nonlinear load	0.645	0.531	0.523	0.359	0.441	0.319
Current phase *a* THD [%]	Balanced load	1.44	1.23	0.75	0.72	0.70	0.68
	Unbalanced load	2.14	2.08	1.86	1.41	1.58	1.37
	Nonlinear load	2.93	2.86	2.60	2.53	2.56	2.23

## Data Availability

Not applicable.

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
