# Peer review of "Comparative Performance Analysis of the DC-AC Converter Control System Based on Linear Robust or Nonlinear PCH Controllers and Reinforcement Learning Agent"

_sensors, 2022, doi:10.3390/s22239535_

Round 1
Reviewer 1 Report
1. Please avoid using bulk citations in the introduction, and evaluate them separately.
2. Introduction requires better state-of-the-art and highlighting the novelty of the paper.
3. What will be the impact of having a storage system on the DC side? for example in Figure 1.
4. Is there any reference for the values of Table 1?
5. Please add more discussions in section 3 to support the equations and figures.
6. There are no discussions and explanations in Figures 18 - 41, please add more discussion to support the results.
Author Response
Dear reviewer, thanks for your recommendations.
- and 2. We made the required changes and we inserted in Introduction the following:
In a top-down approach to the general issues that can be found in a microgrid, we can start with the issues of optimization and forecasting from an economic point of view [1,2], and then analyze the control elements of the main subassemblies of the microgrid, i.e. the DC-DC converter [3,4], DC-AC converter [5,7], battery energy storage system (BESS) [8,9], and last but not least specific connection elements in the case of electric vehicles connected to the microgrid [10].
Naturally, in order to obtain superior control performances, a series of modern types of controllers have been developed and implemented specifically for the control of the main elements of the microgrid described above, including adaptive controllers [12], robust controllers [13-17] in case of significant parametric variations, neuro-fuzzy controllers [18], as well as nonlinear controllers based on the Passivity theory, including nonlinear PCH [19-23].
The microgrid topology discussed in this article and the control objectives are based on a benchmark presented in [16,17,22,23].
The main contributions of this paper can be summarized as follows:
- Presentation, synthesis, and implementation of the robust control algorithm for DC-AC converter control;
- Presentation, synthesis, and implementation of the PCH control algorithm based on the passivity theory for the DC-AC converter control;
- Presentation, synthesis, and implementation of a RL-TD3 agent, by covering the stages of creation, training, testing and validation for each of the PI, robust and PCH controllers;
- Implementation in Matlab/Simulink of the software applications for the calculation of the steady-state error performance indicators and the error ripple of the ud voltage and THD current phase a of the microgrid-to-the-main-grid connection system using a DC-AC converter for the comparative analysis of PI, robust and PCH control systems with or without the RL-TD3 agent.
- The grid-characteristic currents ia, ib, ic, are dictated by the consumers connected to it and represent the input quantities for the robust controller, which will be synthesized using the robust systems theory. This controller will supply the control signals to a PWM generator, and, by driving active MOSFET or IGBT elements in the DC-AC converter, ud voltage will be kept constant, which is the main objective of the control system for the presented benchmark. We specify that in the microgrid topology shown in Figure 1 there is no BESS, precisely in order to follow the benchmark presented. From the point of view of the synthesis of the controllers proposed in this article, the absence or presence of a BESS does not influence the synthesis of these controllers or the performance of these control systems. This is due to the fact that in the currents ia, ib, ic which represent inputs for the controller, there are fluctuations caused by consumers, and possible BESS’, both in stationary regime and in dynamic regime as a result of their connection or disconnection. Moreover [8] presents the control of the main phenomena occurring when there is a BEES, namely their charging or discharging according to certain criteria imposed by the connection to the microgird. These refer to the charging and discharging of the BESS when the voltage at its terminals is lower, respectively higher by a set percentage than the voltage which is intended to be kept constant in the microgrid. These goals are achieved through the use of classical PI-type cascade controllers, where the charging/discharging current of the BESS is regulated in the inner loop and the voltage at the BESS terminals is regulated in the outer loop.
- The values of table 1 are for a benchmark from [16,17,22,23].
The robust controller is presented in Section 2 and is described using equations (1) to (12), and for these are given references for the readers.
The PCH-type is described in the same manner in Section 3 using equations (14) to (31) and, also for the synthesis of this controller is given references for the readers.
- We inserted new comments for Figures 18-41.
“Thus, Figures 18-20 show the time evolution of ud and uq voltages for DC-AC converter control system based on PI controller in the case when the load is balanced, unbalanced or nonlinear. Figures 21-23, for the same types of load variation, show the time evolution of ud and uq voltages for DC-AC converter control system based on PI controller improved by using a RL-TD3 agent. Substantial improvement in control system performance can be observed when using PI control in combination with an RL-TD3 agent.
Figures 24-26 show the time evolution of ud and uq voltages for DC-AC converter control system based on robust controller when the load is balanced, unbalanced or nonlinear. Figure 27-29, for the same types of load variation, show the time evolution of ud and uq voltages for DC-AC converter control system based on robust controller improved by using a RL-TD3 agent. Substantial improvement in control system performance can be observed when using the robust control in combination with an RL-TD3 agent.
Figures 30-32 show the time evolution of ud and uq voltages for DC-AC converter control system based on PCH-type controller in the case when the load is balanced, unbalanced or nonlinear. Figures 33-35, for the same types of load variation, show the time evolution of ud and uq voltages for DC-AC converter control system based on PCH-type controller improved by using a RL-TD3 agent. Substantial improvement in control system performance can be observed when using PCH-type control in combination with an RL-TD3 agent.”
“Figures 36-41 show the FFT analysis and THD for current phase a of the DC-AC converter controller for the types of controllers and load variations presented above. Figures 36 and 37 show FFT analysis and THD for the current on phase a of the DC-AC converter controlled with PI-type controller without/with RL-TD3 agent in the case of balanced, unbalanced or nonlinear type for the load. Figures 38 and 39 show FFT analysis and THD for the current on phase a of the DC-AC converter controlled with robust-type controller without/with RL-TD3 agent in the case of balanced, unbalanced or nonlinear type for the load. Figures 40 and 41 show FFT analysis and THD for the current on phase a of the DC-AC converter controlled with PCH-type controller without/with RL-TD3 agent in the case of balanced, unbalanced or nonlinear type for the load.”

Reviewer 2 Report
The paper is very interesting in the learning machine domain, few ccorrections

Author Response
Dear reviewer, thanks for your recommendations.
1, 2, and 3. The requested corrections were made and inserted into the text of the article.
4. Naturally, in order to obtain superior control performances, a series of modern types of controllers have been developed and implemented specifically for the control of the main elements of the microgrid described above, including adaptive controllers [12], robust controllers [13-17] in case of significant parametric variations, neuro-fuzzy controllers [18], as well as nonlinear controllers based on the Passivity theory, including nonlinear PCH [19-23].
In terms of Machine Learning types, it can be mentioned the RL-TD3 agent [24-27], which can improve the performance of the DC-AC converter control system. The RL-TD3 agent resembles the architecture of an industrial process control system through a very strong analogy in terms of information acquisition and command provision, as well as optimization of an overall quality index. After the phases of training and validation of an RL-TD3 agent, it provides correction signals to the command signals leading to optimized and increased performance of the control system.
The microgrid topology discussed in this article and the control objectives are based on a benchmark presented in [16,17,22,23]. Thus, the performance of DC-AC converter control systems is compared when using PI-type, robust and PCH-type controllers.
